# Aromatic Plant-Based Functional Foods: A Natural Approach to Manage Cardiovascular Diseases

**DOI:** 10.3390/molecules28135130

**Published:** 2023-06-30

**Authors:** Mónica Zuzarte, Henrique Girão, Lígia Salgueiro

**Affiliations:** 1University Coimbra, Coimbra Institute for Clinical and Biomedical Research (iCBR), Faculty of Medicine, 3000-548 Coimbra, Portugal; 2University Coimbra, Center for Innovative Biomedicine and Biotechnology (CIBB), 3000-548 Coimbra, Portugal; 3Clinical Academic Centre of Coimbra (CACC), 3000-354 Coimbra, Portugal; 4University Coimbra, Faculty of Pharmacy, 3000-548 Coimbra, Portugal; 5University Coimbra, Chemical Process Engineering and Forest Products Research Centre (CIEPQPF), Department of Chemical Engineering, Faculty of Sciences and Technology, 3000-548 Coimbra, Portugal

**Keywords:** nutrition, essential oils, fortified foods, enhanced foods, encapsulation, cardioprotection

## Abstract

Aromatic plants and their essential oils have shown beneficial effects on the cardiovascular system and, therefore, are potential raw materials in the development of functional foods. However, despite their undeniable potential, essential oils present several limitations that need to be addressed, such as stability, poor solubility, undesirable sensory effects, and low bioavailability. The present review provides a current state-of-the-art on the effects of volatile extracts obtained from aromatic plants on the cardiovascular system and focuses on major challenges that need to be addressed to increase their use in food products. Moreover, strategies underway to overcome these limitations are pointed out, thus anticipating a great appreciation of these extracts in the functional food industry.

## 1. Introduction

Functional foods are generally described as foods that offer health benefits beyond their nutritional value [1]. This concept appeared in 1980 in Japan and rapidly extended to other parts of the world, covering a variety of products including whole, processed, fortified, enriched, or enhanced foods. Nevertheless, according to a recent definition, only novel foods that have been formulated to contain substances or live microorganisms with health-enhancing or disease-preventing value, at a safe and sufficiently high concentration, should be considered a functional food [2]. Despite the lack of consensus in the used terminology, the current market of functional foods is blooming, attaining USD 280.7 billion in 2021 and is expected to expand at a compound annual growth rate (CAGR) of 8.5% until 2030. The major functional food claims include sports nutrition, weight management, immunity boost, digestive health, clinical nutrition, and cardio health [3]. Since cardiovascular diseases (CVDs) have a major impact on the health sector, being the leading cause of mortality worldwide, the market of healthy diets, including functional foods is expected to propel. Currently, several functional foods, such as soybeans, oats, psyllium, flaxseed, garlic, tea, fish, grapes, nuts, and stanol- and sterol-ester enhanced margarine have been pointed out as potentially beneficial in the prevention and treatment of CVDs. It seems that the ingestion of adequate amounts of these foods decreases the risk of CVDs due to their capacity to lower blood lipid levels, improve arterial compliance, reduce low-density lipoprotein oxidation, decrease plaque formation, scavenge free radicals, and inhibit platelet aggregation [4].

In addition to these natural products, aromatic plants play a very interesting role as they can be used as a whole food, being included in several dishes, such as soups and salads or used as spices and condiments (e.g., basil, lavender, mint, oregano, rosemary, sage, thyme, among others), thus allowing for the reduction in the consumption of salt and fat, and overall contributing to good dietary practices [5]. Moreover, several herbs are traditionally used to prevent and treat CVDs with several studies validating their cardioprotective potential. For example, *Lavandula angustifolia* showed a cardioprotective in vivo effect against isoproterenol-induced myocardial infraction, by decreasing tissue damage and strengthening the myocardium membrane [6]. The bioactive potential attributed to aromatic species is due to the presence of secondary compounds, such as volatile terpenes. Indeed, some of these compounds have also demonstrated cardioprotective potential, such as carvacrol, present in oregano essential oil that was able to suppress myocardial ischemic damage in vivo by diminishing the infarct size and myocardial enzymes [7]. Moreover, citronellol, present in several essential oils, showed anti-hypertensive effects [8] and limonene, found in high amounts in citrus fruits, ameliorated cardiac injury induced by CCl4 intoxication due to its antioxidant and anti-inflammatory potential [9].

However, to achieve health benefits, significantly higher quantities of plants and their bioactive metabolites are required and, therefore, the addition of plant extracts, namely, essential oils, to foods is a suitable strategy to be considered. Nevertheless, several features of these volatile extracts, such as sensory properties, safety, stability, as well as regulatory issues need to be addressed to comply with food ingredient requirements [10]. Moreover, innovative strategies that enable a controlled release of the essential oils have been proposed to overcome some of these limitations. The present review tackles these points by presenting a general overview on essential oils produced by aromatic plants, outlining their beneficial effects on the cardiovascular system, referring to their current use in food products, pointing out major challenges that need to be overcome to include these volatile extracts in other functional foods, and addressing current solutions and future directions.

## 2. Overview on Aromatic Plants and Their Volatile Extracts

Aromatic plants are defined as plants that produce and accumulate volatile compounds in secretory structures, such as trichomes, secretory cells or secretory ducts, or cavities. The volatile extracts obtained from these plants are named essential oils and, according to the International Standard Organization on Essential Oils [11] and the European Pharmacopoeia [12], are isolated from the plant raw material by hydrodistillation, steam distillation or dry distillation, or by a suitable mechanical process in the case of *Citrus* fruits.

Essential oils play important roles in plants ecophysiology, namely, plant defense, environmental adaptation, and polinization. Moreover, humankind has discovered several applications, with the food industry standing out as one of the most relevant, as several essential oils are considered safe for consumer use and have a generally recognized safe (GRAS) status by the Food and Drug Administration (FDA). Furthermore, several essential oil compounds are included in the EU list of flavoring substances that can be used in food, approved by the European Commission (Figure 1). However, the regulation of essential oils for human consumption varies according to the intended mode for administration, although good manufacturing practices are always required [10].

Essential oils are usually complex mixtures of several compounds of low molecular weight, mainly monoterpenes and sesquiterpenes, although in some cases non-terpenic compounds, such as phenylpropanoids, as well as sulphur- and nitrogen-containing compounds are quite relevant [13]. As several factors, including physiological, environmental, and genetic factors influence the composition of essential oils, specific guidelines need to be considered to attain high quality standards for commercialization. Indeed, chemical variability in aromatic plants occurs quite often and can influence the bioactivity of essential oils. Therefore, the pharmacopoeia monographs include a chromatographic profile with the minimum and maximum limits of a certain number of compounds to guarantee the quality of these oils and, consequently, their efficacy and safety. Furthermore, guidelines for Good Agricultural and Wild Collection Practices of Medicinal and Aromatic (Culinary) plants (GACP-MAP) are available for the cultivation, wild collection, and primary processing practices of these plants, and their derivatives are traded and used in the European Union [14].

Around 300 essential oils have been marketed in flavor and fragrance products with the main trade oils attaining over 1000 t/year. The essential oils most commonly used in food products include anise, basil, bergamot, chamomile, green and black cumin, coriander, garlic, lavender, lemon, lemon balm, myrtle, neroli, rosemary, orange, peppermint, pine, thyme and yarrow, as recently reviewed [15]. The global flavors and fragrances market is expected to grow with a very recent report predicting an increase of up to USD 44.6 billion by 2030. This escalation seems to be related with the rising product demand from the nutraceutical, dietary supplements, pharmaceuticals, and food processing industries [3].

In addition to fragrance and flavoring properties, essential oils are quite relevant in food safety. Indeed, their antimicrobial and antioxidant properties enable their use, for example, in food packaging to control spoiling agents, thus enhancing the shelf life of food [16]. Essential oils can be found in a variety of food products as detailed in the next section and, as their use in functional foods is rather promising, several novel products are expected to be developed in the following years (Figure 1). Moreover, several studies have pointed out other beneficial effects of these extracts on human health including anti-inflammatory, neuroprotective, and cardioprotective properties, as reviewed in [17,18,19], justifying their use in dietary products. However, the direct use of essential oils in foods can decrease their functional properties due to uncontrolled release, poor solubility, undesirable sensory effects of some oils, and low bioavailability [20]. Moreover, some bioactive compounds can be potentially lost or reduced during food processing. Therefore, effective strategies that enable a controlled release of these extracts, such as encapsulation technologies have been developed, as addressed in Section 4. Despite these challenges, the incorporation of essential oils into foods is in line with green consumerism and allows for the use of clean labels on foods, making them a very tempting raw material for the food sector.

## 3. Potential of Aromatic Plants and Volatile Extracts on the Cardiovascular System

Cardiovascular diseases (CVDs) greatly impact health care systems as they continue to be the leading cause of mortality worldwide, accounting for 32% of total deaths [21]. This growth correlates with the increase in associated risk factors, with ageing standing out as one of the most relevant. Indeed, it is estimated that by 2030, 20% of the world’s population will be more than 65 years old. Ageing together with genetic predisposition, socioeconomic status, and ethnicity are examples of non-modifiable risk factors that have been associated with CVDs incidence. In addition to these risk factors, sedentary life style, obesity, unhealthy diet and habits, such as smoking and alcohol abuse, hypertension, dyslipidemia, and diabetes, have a great influence on CVDs incidence and progression. Due to their modifiable nature, these factors are included in the WHO’s target list to be reduced by 2025 [22].

Therapeutic options for CVDs are also available and these primarily target associated risk factors, avoiding disease progression and further complications. However, their efficacy is compromised by low patience adherence (~40%), mainly due to undesirable side-effects [23]. Therefore, in addition to strategies for the reduction in modifiable risk factors, new effective preventive and therapeutic approaches are in urgent need. In this context, aromatic plants have deserved special attention as herbal medicines are used by ca. 80% of the world´s population in basic health care and several of these plants are part of the Mediterranean diet with proven benefits on the cardiovascular systems [24]. The beneficial effects of this diet have been associated with the consumption of fruit, vegetables, spices, garlic, and onions [25] and the preventive/therapeutic potential of aromatic plants is mainly attributed to the presence of secondary metabolites, such as terpenes, found in plant´s volatile extracts. In addition, in the last years, researchers have focused their attention on the cardioprotective potential of aromatic plants and their volatile extracts, as highlighted in recent reviews [26,27]. Overall, the majority of the studies, carried out to date, include in vitro and in vivo approaches that emphasize the capacity of essential oils to reduce the damaging outcomes of CVDs major risk factors, such as hypertension, diabetes, and dyslipidemia, among others.

Studies on the anti-hypertensive effect of essential oils are the most reported and typically normotensive and hypertensive animal models are used. In these models, the effect of the oils following hypertension induction with deoxycorticosterone acetate (DOCA)-salt administration or nephrectomy is assessed. Generally, heart parameters are measured, such as mean arterial pressure, systolic pressure, diastolic pressure, and heart rate.

To access the antidiabetic and antidyslipidemic potential of essential oils, normally rat models of diabetes induced by streptozotocin or rats submitted to high fat or high cholesterol diets are used, respectively. Markers of altered lipid and cholesterol metabolism are assessed, namely, total triglycerides, total cholesterol, and low- and high-density lipoproteins.

In less extent, the effect of essential oils on antiplatelet aggregation has also been shown. In these cases, the pre-clinical model of thromboembolism is used as it enables the assessment of death prevention and paralysis events. Other beneficial effects of essential oils on the cardiovascular system have also been assessed, such as their in vitro effect on calcium channel modulation.

Some of the referred essential oils are rich in phenylpropanoids, such as eugenol (*Ocimum gratissimum* and *Syzygium aromaticum*), anethole (*Foeniculum vulgare*), estragole and methyl eugenol (*Croton zehntneri*) that have also shown very effective effects on the cardiovascular system, as reviewed in [19]. Indeed, eugenol has shown vasorelaxant properties [28,29] and was able to ameliorate insulin resistance, oxidative stress, and inflammation in high fat-diet/streptozotocin-induced diabetic rats [30]; anethole showed an antithrombotic effect that was associated with its antiplatelet activity, clot destabilizing effect, and vasorelaxant action [31] and methyl eugenol elicited hypotension related to vascular relaxation in normotensive rats [32]. Other relevant compounds in the cardiovascular field are terpenes, such as limonene, a major compound of the essential oils of several citrus fruits. Limonene has shown bradycardic and antiarrhythmic effects in rats [33] and was able to ameliorate cardiac injury induced by carbon tetrachloride intoxication due to its antioxidant and anti-inflammatory potential [9]. Moreover, 1,8-cineole, a compound present in varying concentrations in several essential oils and a major compound, for example, in *Alpinia zerumbert*, *Salvia officinalis,* and *Ocimum gratissimum* essential oils is also quite promising. A recent study pointed out its anti-hypertrophic effect and improved right ventricle function in the rat monocrotaline-induced pulmonary arterial hypertension model. This compound was able to restore gap junction protein connexin43 distribution at the intercalated disks and mitochondrial functionality, suggesting its potential to preserve cardiac cell-to-cell communication and bioenergetics [34]. Interestingly, in another study, the combined application of 1,8-cineole and β-caryophyllene, a major compound of *Pogostemon elsholtzioides*, synergistically reversed cardiac hypertrophy in isoprenaline-induced H9c2 cells and mice [35]. Piperitenone oxide, present in high amounts in *Mentha* × *villosa* essential oils, elicited immediate and dose-dependent decrease in mean arterial pressure and heart rate in normotensive rats [36]. Another relevant compound is caryophyllene oxide present in *Artemia campestres* essential oils. This compound was able to inhibit potassium currents, thus reducing the negative inotropic effect related to the blockade of calcium channels and justifying its use, for example, in arrhythmias [37]. The effectiveness of these compounds suggests that, in some cases, the major compounds found in essential oils (Figure 2) are responsible for their bioactivities. However, as many essential oils tend to be more active than their isolated compounds, it is likely that synergistic effects between different compounds may occur.

In addition to in vivo animal studies, clinical trials have been performed, highlighting the potential of essential oils in the cardiovascular field, namely, as anti-hypertensive agents. Moreover, although in the majority of the cases the essential oils were inhaled, some studies refer to their ingestion, reinforcing a potential use in food products. Table 1 compiles these studies by gathering information on the cardiovascular condition/feature assessed, the essential oil´s trade name and chemical composition if provided, plant species scientific name (when not referred, the presumed species is indicated with an asterisk) and family, number of individuals enrolled in the trial, type of clinical intervention, dose (when referred), and main outcomes. When a blend of essential oils was used, this information is also provided.

Despite the relevance of these clinical trials that pave the way to an effective translation, it is important to mention that many of these studies do not refer to the chemical composition of essential oils. This is quite an important aspect to consider bearing in mind the variability of essential oils that can compromise their use in the clinical setting and in other applications, such as the food industry. Therefore, additional studies with standardized essential oils should be performed to avoid variability and guarantee quality, safety, and efficacy issues. In addition, the scientific name of the plant from where the essential oil was obtained is missing in the majority of the studies. Most probably, commercial essential oils were used; however, stating the plant´s scientific name is of utmost relevance to avoid confusion as common names are highly variable. For example, lavender is a common name for any of the 39 known species of flowering plants of the genus *Lavandula* [47].

Overall, in the cardiovascular field, lavender essential oil is one of the most studied, being considered in several clinical trials. Although the scientific name of the plant is not referred, it is assumed that the oil tested is that of *Lavandula angustifolia*, as this species is the most used worldwide. According to the European Pharmacopoeia, the main compounds of *Lavandula angustifolia* essential oil are linalool (20–45%) and linalyl acetate (25–46%) [12]. It is likely that the positive effects observed for the essential oil on the cardiovascular system rely on these compounds. Indeed, linalool assessed alone was able to reduce blood pressure without changing the heart rate of hypertensive rats [48] and showed cardioprotective effects in the isoproterenol-induced myocardial infarction rat model by improving the oxidative condition and abolishing both apoptotic and inflammatory responses [49]. Moreover, linalyl acetate showed in vivo cardiovascular effects by recovering cell damage and cardiovascular changes induced by acute nicotine [50], exerting an anti-hypertensive effect that prevented hypertension-related ischemic injury [51] and by restoring acetylcholine-induced vasorelaxation, blood pressure, and heart rate in diabetic rats exposed to chronic immobilization stress [52].

In addition to the direct effects of essential oils on major risk factors directly associated with CVDs, several clinical trials have validated other effects that are quite relevant in cardiac patients, such as stress and anxiety related to their cardiovascular condition or pain, fatigue, and anxiety due to cardiac interventions. For example, rose essential oil reduced the anxiety in patients with acute myocardial infarction [53] and the inhalation of peppermint essential oil significantly reduced anxiety in patients with acute coronary syndrome in the emergency department [54]. Moreover, inhalation of lemon balm essential oil decreased both acute stress, threat perception, and pain intensity in patients with acute coronary syndrome in the emergency department [41]. In addition, aromatherapy with peppermint and lavender essential oils were effective in reducing cardiac patient’s fatigue.

Bearing in mind the current state-of-the-art on the beneficial effects of essential oils on the cardiovascular system and their highly valued interesting features in the food industry, it is plausible that their use as raw materials in the development of functional foods will bloom in the following years (Figure 1). Furthermore, the use of essential oil in the field of CVDs prevention is expected as inflammation and cardiovascular diseases are highly linked and essential oils are known for their effective anti-inflammatory potential [55,56].

## 4. Challenges in the Use of Essential Oils in Functional Foods

The potential of essential oils in the food industry is undeniable. In fact, many applications are currently explored, with essential oils being used in foods as antimicrobial agents, food additives, food enhancers, flavoring agents and adjuvants, food contact substances, preservatives or processing aids, as recently reviewed in [10]. Moreover, their use within food packaging material is quite common. Particularly in the last years, the use of these volatile compounds as biopreservatives has received increasing attention mainly due to consumers’ concerns toward the use of synthetic preservatives. Indeed, microbial contaminants affect crops, foods, commodities, and raw material and are a relevant source of mycotoxins that can cause severe health concerns [57]. Importantly, following ingestion, these compounds are able to suppress the immune system, induce mutations, cancer, teratogenic effects, and infertility [58]. Although synthetic fungicides may be used to control these contaminations, their continuous application disrupts ecosystems leading to disease outbreaks, development of resistant strains, toxicity to non-target organisms, and environmental concerns [59]. Due to their antimicrobial properties, essential oils can be used as natural alternatives, being currently applied in a variety of food products, such as meat, seafood, fruits and vegetables, dairy, and cereal-based products [60].

In addition to these applications, researchers have ascribed to essential oils other health promoting properties, namely, in the cardiovascular field that justify the development of new aromatic plant-containing products, namely, functional foods with preventive/therapeutic potential. In addition to the reported beneficial effects primarily in major associated risk factors referred in Section 3, these extracts may be considered candidates for prophylaxis of CVDs Indeed, bearing in mind the role of inflammation in the pathophysiology of CVDs and the well-known anti-inflammatory potential of several essential oils, it seems plausible that the consumption of these compounds over a period of time could be an effective preventive strategy [61]. Moreover, the use of aromatic plants and their extracts as prebiotics may be considered prophylactic in this context as gastrointestinal bacterial dysbiosis also plays a very relevant role in systemic inflammation.

However, to consider the use of aromatic plants and their essential oils in food products, several issues need to considered, namely, regulatory requirements, safety concerns, sensory properties, and essential oil´s stability in foods that need to be further processed. Furthermore, these extracts gather additional particularities, such as volatility and hydrophobicity, making their use even more challenging in products that are ingested. Importantly, oral bioavailability is crucial but accurate studies on pharmacokinetics and bioavailability of essential oils are quite sparse. It has been suggested that these compounds are rapidly absorbed following ingestion, in the small intestine, and are then metabolized or eliminated by the kidneys in the form of phase-II conjugates, mainly glucuronides [62]. However, detailed information on their absorption, metabolism, distribution, and elimination is still required.

Bearing in mind the lipophilicity of essential oils, they can be dissolved in food lipid matrices, be incorporated in the lipid phase of emulsified products, or a surfactant with high hydrophilic/lipophilic balance can be used to enable their micellar dispersion into aqueous food products [15]. Considering their volatility, heating processes should be avoided to not compromise stability. Another relevant aspect to consider is essential oil safety. Although clinical studies, carried out to date, showed negligible side effects of essential oils that were well-tolerated by patients, interactions with prescription medications, namely, due to pharmacokinetic or pharmacodynamic interactions can occur. Moreover, as the mode of action of many essential oils is not fully elucidated, the cytotoxicity of these extracts on non-target cells should also be considered. Furthermore, essential oils may have a negative impact on food´s sensory properties due to their strong aroma and flavor that can cause undesirable organoleptic properties and compromise compliance.

To reduce these negative impacts, different nanoencapsulation strategies including emulsions, lipid nanoparticles, biopolymeric nanoparticles, clay-based nanoparticles, and inclusion complexes, have been applied to prevent unwanted degradation and enhance the efficacy of essential oils with several applications, particularly in the food sector [63]. Nanoencapsulation is generally referred to the use of a carrier with less than 1-micron, but since the size of the delivery system influences the surface area and dispersion of the essential oil into the food matrices, a size less of than 100 nm is recommended [64]. These nanocapsules can be added to the food during production, processing, packaging, and security [65]. Indeed, encapsulation techniques are quite interesting in the food sector as they protect essential oils from degradation, increase their solubility in a hydrophilic environment, mask their strong aroma, avoid interactions with other food components, and enable a targeted administration that could decrease the necessary dose [66]. For example, developing gastro-resistant formulations enables the maximization of the efficacy of essential oils, as more amounts are absorbed in the intestine, thus enabling an effective dose reduction. Moreover, nanoencapsulated essential oils have improved food shelf-life and preservation [67,68] and can be added directly to foods, or into food matrices in the form of emulsions or be used in food packaging. For example, edible coatings have been successfully applied as they lengthen the shelf-life of the food product and increase its value. Importantly, the incorporation of essential oils in these coatings can add additional properties, such as antioxidant and/or antimicrobial characteristics, thus increasing even more the product´s value [69].

More recently, studies on the use of plant nanovesicles as drug-delivery platforms have emerged [70]. These vesicles have been pointed out as promising encapsulation alternatives to mammalian vesicles and synthetic carriers due to their nontoxic and nonimmunogenic character. In a recent review, it was postulated that these vesicles could be an excellent choice to encapsulate bioactive essential oils and effectively deliver them to the target organs [71]. Moreover, in the context of aromatic plant-based functional foods for cardiovascular diseases, these vesicles are an excellent choice. Importantly, plant vesicles can be produced in a large-scale from various plant parts, enabling less costly industrial applications that could add value to waste products and boost circular economy. Another relevant aspect to consider is the use of functionalized vesicles with heart-targeting strategies, such as homing peptides [72] to effectively deliver the encapsulated vesicles to the target organ, ultimately leading to dose reduction and patient compliance.

Overall, it seems undeniable that the use of essential oils in the food industry, and particularly, in the development of next-generation functional foods will resort to the use of delivery nanoplatforms to overcome the referred challenges.

## 5. Conclusions and Future Perspectives

Aromatic plants and their essential oils have long been used by the food industry mainly as natural preservatives. However, their beneficial cardiovascular potential points out new directions with the functional food sector standing out as one of the most promising. Indeed, several essential oils have shown promising effects mainly on CVDs major risk factors, such as hypertension, diabetes, and dyslipidaemia. Importantly, clinical trials have validated the potential of lavender, ylang-ylang, manjoram, neroli, bergamot, lemon balm, green cumin, cumin, nigella, and Damask rose essential oils in this field. Nevertheless, to enable the development of effective food products, some essential oil features, such as low water solubility, high volatility, strong odor, and distinctive flavor need to be taken into account. In recent years, several approaches have emerged to overcome these challenges, with encapsulation strategies being developed and enabling an increase in the bioavailability and chemical stability of these compounds. More recently, plant nanovesicles have emerged as a promising alternative for essential oil encapsulation due to their nontoxicity and nonimmunogenic potential. Moreover, their mass production can be attained with fewer costs [71] and their use could increase the bioavailability of essential oils and improve their chemical stability, thus reducing volatility and toxicity. In addition, it seems that these vesicles have targeting properties that could be explored to direct essential oils to a specific organ, improving selectivity, delivery, efficacy, safety, and ultimately yielding a reduction in the essential oil dose needed. However, regarding stability under the harsh conditions of the gastrointestinal tract, contradictory results have been published, with some studies pointing out plant vesicles´ resistance in terms of size, surface distribution, and surface charge [73], whereas others report modifications in these features [74]. Nevertheless, this possible drawback can be overcome by enteric coating to retain vesicle integrity and enable the release of essential oils where they are expected to be absorbed or act [71]. Plant nanovesicles have been applied as oral delivery systems in different pathologies, such as colitis, bowel, liver, neurodegenerative, and metabolic diseases [74,75]. These applications could be extended to CVDs and the therapeutic effect synergistically complemented with the essential oil administration, thus developing an interesting strategy for the functional food sector.

Overall, essential oils gather relevant features that enable several applications in the food industry and the development of further health-oriented functional foods, but impose some challenges that will need to be addressed to guarantee the effectiveness of new plant-based functional foods. Overall, the current state-of-the-art on the health-promoting cardiovascular effects of essential oils and the innovative advances that enable the safe and desirable ingestion of these extracts are definitely paving the way for the development of new functional foods in the next years.

## Figures and Tables

**Figure 1 molecules-28-05130-f001:**
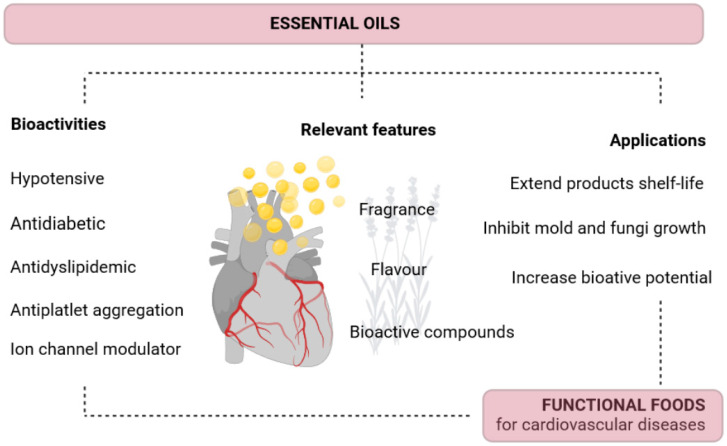
The potential of essential oils to develop functional foods directed toward cardiovascular disease prevention and treatment. Created with Biorender.

**Figure 2 molecules-28-05130-f002:**
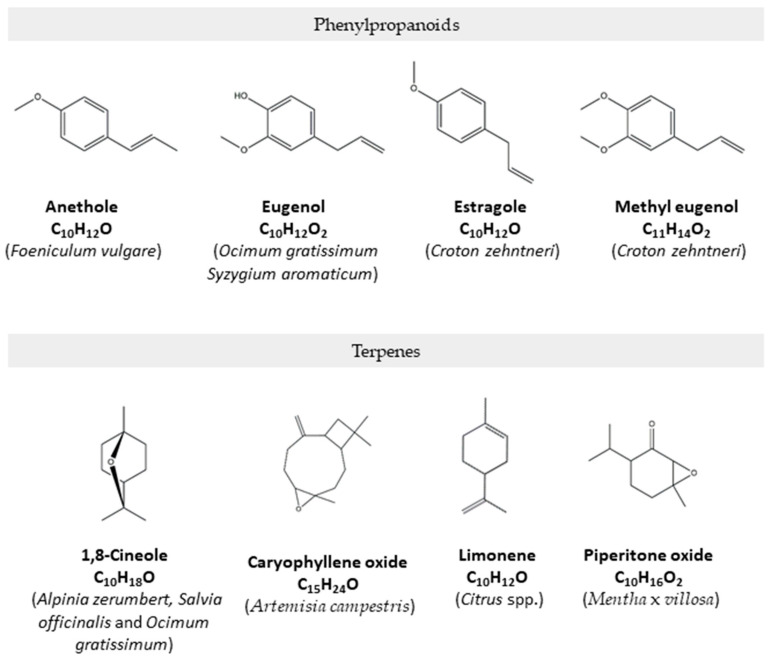
Main cardioprotective compounds found in essential oils.

**Table 1 molecules-28-05130-t001:** Clinical trials on the effects of essential oils on the cardiovascular system.

Cardiovascular Condition	Essential Oil Trade Name (Chemical Composition)	Plant Scientific Name (Family)	Type and Number of Participants	Type of Intervention	Dose and Treatment	Outcomes	Ref.
Hypertension	Blend: Lavender, marjoram, ylang-ylang, and neroli (20:10:15:2)	*Lavandula angustifolia* * (Lamiaceae); *Origanum majorana ** (Lamiaceae); *Cananga odorata ** (Annonaceae); *Citrus aurantium* * (Rutaceae)	Middle-aged women with hypertension; n = 99	Aroma massage and aroma body cream	Five sessions, during 4 weeks [lavender, marjoram, ylang-ylang, and neroli blended in 20:10:15:2 ratio and diluted to 3%]	**↓** Blood pressure and **↑** quality of sleep	[38]
	Blend: Lavender, ylang-ylang, and bergamot (5:3:2)	*Lavandula angustifolia* * (Lamiaceae); *Cananga odorata ** (Annonaceae); *Citrus bergamia* (Rutaceae)	Hypertensive individuals; n = 52	Aromatherapy	Daily for 4 weeks	**↓** Blood pressure and stress responses	[39]
	Blend: Lemon, lavender, and ylang-ylang (2:2:1)	*Citrus limonum* (Rutaceae); *Lavandula angustifolia* (Lamiaceae); *Cananga odorata* (Annonaceae)	Essential hypertensive patients; n = 42	Aromatherapy	2 min per inhalation, two times per day for 3 weeks [lemon, lavender, and ylang-ylang blended in a 2:2:1 ratio]	**↓** Systolic blood pressure	[40]
	lemon balm	*Melissa officinalis* (Lamiaceae)	Patients with acute coronary syndrome upon admission to the emergency department; n = 72	Aromatherapy	Two drops in two aromatherapy phases for 10 min with 90-min interval	Regulation of mean arterial pressure and heart rate	[41]
Diabetes	green cumin [major compounds: Cumin aldehyde, thymoquinone, p-cymene, γ -phellandrene, limonene, and myrcene]	*Cuminum cyminum* (Apiaceae)	Patients with diabetes type II; n = 99	Ingestion	50 or 100 mg/day for 8 weeks	**↓** Glycemic indices, insulin resistance, and serum inflammatory factors	[42]
	cumin [major compounds: Cumin aldehyde (41.9%), γ-terpinene (16.5%), ρ-cymene (16.2%), and β-pinene (10.9%)]	*Cuminum cyminum* (Apiaceae)	Pre-diabetic patients; n = 64	Ingestion	One soft gel (75 mg of cumin EO) per day for 10 weeks	Ameliorated insulin function and lipid profile, and anthropometric indices	[43]
	black seed	*Nigella sativa* (Ranunculaceae)	Type II diabetic patients; n = 70	Ingestion	2.5 mL, two times a day after meals for 3 months	**↓** Glucose a, HbA1c, and BMI	[44]
		*Nigella sativa* (Ranunculaceae)	Diabetic patients undergoing hemodialysis; n = 46	Ingestion	One capsule (with 2 g of oil) per day after hemodialysis and apart from meals	**↑** Levels of SOD, MDA, TAC, hs-CRP, HbA1c, and FBS	[45]
Cardiovascular disease	Damask Rose	*Rosa damascena* (Rosaceae)	Patients undergoing coronary angiography; n = 98	Inhalation before coronary angiography	Five drops of 40% EO for 20 min	**↓** Hemodynamic parameters	[46]

HbA1c—hemoglobin A1c; BMI—body mass index; EO—essential oils; FBS—fasting blood sugar; HbA1c—glycosylated hemoglobin; hs-CRP—C-reactive protein; MDA—malondialdehyde; SOD—superoxide dismutase; TAC—total antioxidant capacity; high-sensitivity: *—presumed plant species; ↓—decrease; ↑—increase.

## Data Availability

Data are contained within the article.

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
