# Peer review of "Aromatic Plant-Based Functional Foods: A Natural Approach to Manage Cardiovascular Diseases"

_molecules, 2023, doi:10.3390/molecules28135130_

Round 1
Reviewer 1 Report
The review paper is well structured. It is of interest to the scientific community. My opinion is to accept after minor revision. See attached file.

Author Response
We appreciate your suggestions and have accepted all the alterations pointed out in the pdf.
Reviewer 2 Report
The article “Aromatic Plant-based Functional Foods: a natural approach to manage cardiovascular diseases” reviews published a paper about essential oils' potential in preventing cardiovascular disease.
In my opinion, in the presented form this paper is not suitable for a high-ranked journal such as Molecules. In my opinion, there is not enough novelty in this manuscript, there are too many similarities with previous work (The Role of Essential Oils and Their Main Compounds in the Management of Cardiovascular Disease Risk Factors. Molecules 2021;26:3506. https://doi.org/10.3390/molecules26123506). At least 30 references (26,27,31-54, 60, 61,63,66,72 ) reviewed in this manuscript were detailed reviewed in previous work of the same authors.
To improve this manuscript part about recent clinical trials, essential oil encapsulation and application in the food industry, as well as strategies to overcome limits of application in the food industry ) should be more detailed reviewed.
Also, the following queries should be resolved.
1. The sentence in line 53 should be checked and rephrased:
The bioactive potential of aromatic species is many times due to the presence of volatile compounds.
2. In vivo and in vitro should be italic (for example in line 55: oregano essential oil that was able to suppress myocardial ischemic damage in vivo...)
3. Plant nanovesicles as a promising alternative for essential oil encapsulation is mentioned in the conclusion for the first time. It should be mentioned and explained in detail in section 4.
Author Response
To improve this manuscript part about recent clinical trials, essential oil encapsulation and application in the food industry, as well as strategies to overcome limits of application in the food industry should be more detailed reviewed.
We acknowledge the reviewers concerns on the similarities with previous reviews and appreciate its suggestions. We have re-structured our review and included additional information in order to provide more novelty and include information not reviewed before.
Also, the following queries should be resolved.
- The sentence in line 53 should be checked and rephrased:
The bioactive potential of aromatic species is many times due to the presence of volatile compounds.
The sentence was rephrased, as suggested by the reviewer.
- In vivo and in vitro should be italic (for example in line 55: oregano essential oil that was able to suppress myocardial ischemic damage in vivo...)
As suggested all the expressions in vivo and in vitro were written in italic.
- Plant nanovesicles as a promising alternative for essential oil encapsulation is mentioned in the conclusion for the first time. It should be mentioned and explained in detail in section 4.
We acknowledge the reviewers´ comment and provide a more detailed explanation in this type of encapsulation in section 4.
Reviewer 3 Report
Major revisions:
I would recommend to change the title. According to table 1, some of the bioactive effects were detemined in body creams or aromatherapy, not in foods as the title states.
Moreover, most of the information of the manuscript aims to the effect of the bioactive compound, but no imformation is provided regarding their food use. Are any of this effects evaluated on food systems? Which?. Please expand this information. Besides, the effect of food processing condition should also be included.
Minor observations:
Table 1. Include the amount of essential oil that the participants recive during the trearment. The amount should be given in mg in all the cases.
Author Response
I would recommend to change the title. According to table 1, some of the bioactive effects were determined in body creams or aromatherapy, not in foods as the title states.
We appreciate the sugestion; however, several modifications were made to the paper including more focus on the use of these compounds in food and, therefore, if possible we would like to maintain the original title.
Moreover, most of the information of the manuscript aims to the effect of the bioactive compound, but no information is provided regarding their food use. Are any of this effects evaluated on food systems? Which?. Please expand this information. Besides, the effect of food processing condition should also be included.
We acknowledge this relevant point raised by the reviewer and have put an effort in adding more information regarding the food use of essential oils as well as related information.
Minor observations:
Table 1. Include the amount of essential oil that the participants receive during the treatment. The amount should be given in mg in all the cases.
When mentioned in the study, the amount of essential oil that the participants received was mentioned. The units used were those referred in the original study as we are not able to convert mL to mg without density information and some studies refer to the use of drops and not the exact volume.
Reviewer 4 Report
The manuscript deals with the possible applications of essential oils ina managing cardiovascular diseases (CVDs). It provides critical opinion, based on scientific findings. The manuscript is very well planned, organized and written. The English language is satisfactory. There are few comments below on some things that should be improved.
Comments:
Author′s addresses: Do not use abbreviated term “Univ“, use “University Coimbra“
Abstract: last sentence (lines 19, 20, 21) is very hard to understand, please rearrange it or divide it into two sentences.
Keywords: Instead of “fortified foods”, use “functional foods”; delete “” ; add “cardiovascular diseases”
Main body of the manuscript:
Lines 31-35: It seems that the authors reversed references 3 and 4. It should be corrected. Also, there is written that reference 4 is assessed on January 2022, and the date of the online text is February 15, 2023. You must correct the assessment date and also change the value for the functional food market growth by 2027, in the reference 4 it says $354.96 billion.
Furtrher, in this part of the manuscript, you use currency US$, and later, for example in line 104, you use USD. Be consistent, I suggest to use USD.
Line 164: spell check, full stop and coma after “in addition”
Lines 253-254: spell check, percentages and brackets
The biggest complaint is that there are so many general points, referring to essential oil itself, not referring to its role in cardiovascular diseases. For example, lines 312-327 are discussing encapsulation of essential oils, but from general aspect (mainly food and fragrance safety), not at all in the view of cardiovascular diseases. The authors should improve this throughout the whole manuscript. There need to be added important aspects for CVDs.
Table 1: very well summarized references – essential oils, treatments, outcomes.
Figures: My suggestion is to include two more figures, one with the most important aromatic plants used for CVDs, and the second one with the chemical structures of the most important aromatic compounds from these plants, or essential oils important for CVDs, in general. Also, it could be one picture which combines these two segments.
Author Response
The manuscript deals with the possible applications of essential oils in managing cardiovascular diseases (CVDs). It provides critical opinion, based on scientific findings. The manuscript is very well planned, organized and written. The English language is satisfactory. There are few comments below on some things that should be improved.
Comments:
Author′s addresses: Do not use abbreviated term “Univ“, use “University Coimbra“
The abbreviation used is a requirement of our Institution to be mentioned in scientific papers. If this is not an issue according to the journal´s policy, we would like to maintain the rules required by our Institution.
Abstract: last sentence (lines 19, 20, 21) is very hard to understand, please rearrange it or divide it into two sentences.
We acknowledge the suggestion and have rephrased the sentence to a simpler version.
Keywords: Instead of “fortified foods”, use “functional foods”; delete “” ; add “cardiovascular diseases”
We truly appreciate the suggestion. The term ‘fortified food’ was included as it is a type of functional food highly relevant in the food sector; the term ‘functional food’ and ‘cardiovascular disease’, in our opinion, should be avoided as keywords, as they already appear in the title. We have included ‘cardioprotection’ instead.
Main body of the manuscript:
Lines 31-35: It seems that the authors reversed references 3 and 4. It should be corrected. Also, there is written that reference 4 is assessed on January 2022, and the date of the online text is February 15, 2023. You must correct the assessment date and also change the value for the functional food market growth by 2027, in the reference 4 it says $354.96 billion.
This paragraph was rephrased and only one reference included in order to avoid confusion with numbers. The expected compound annual growth rate value was provided instead of the predicted market value.
Further, in this part of the manuscript, you use currency US$, and later, for example in line 104, you use USD. Be consistent, I suggest to use USD.
We agree with the suggestion and have modified US$ to USD
Line 164: spell check, full stop and coma after “in addition”
Thank you. This mistake was corrected
Lines 253-254: spell check, percentages and brackets
Thank you. This mistake was corrected
The biggest complaint is that there are so many general points, referring to essential oil itself, not referring to its role in cardiovascular diseases. For example, lines 312-327 are discussing encapsulation of essential oils, but from general aspect (mainly food and fragrance safety), not at all in the view of cardiovascular diseases. The authors should improve this throughout the whole manuscript. There need to be added important aspects for CVDs.
We appreciate the suggestion and provided more discussion on this topic.
Table 1: very well summarized references – essential oils, treatments, outcomes.
Thank so much for the kind comment.
Figures: My suggestion is to include two more figures, one with the most important aromatic plants used for CVDs, and the second one with the chemical structures of the most important aromatic compounds from these plants, or essential oils important for CVDs, in general. Also, it could be one picture which combines these two segments.
We appreciate the suggestion. A new figure with the suggestions provided by the reviewer was included.